# Evaluation of Collagen Alterations in Early Precursor Lesions of High Grade Serous Ovarian Cancer by Second Harmonic Generation Microscopy and Mass Spectrometry

**DOI:** 10.3390/cancers13112794

**Published:** 2021-06-04

**Authors:** Kristal L. Gant, Alexander N. Jambor, Zihui Li, Eric C. Rentchler, Paul Weisman, Lingjun Li, Manish S. Patankar, Paul J. Campagnola

**Affiliations:** 1Department of Obstetrics and Gynecology, University of Wisconsin-Madison, Madison, WI 53706, USA; kgant@wisc.edu; 2Department of Biomedical Engineering, University of Wisconsin-Madison, Madison, WI 53706, USA; ajambor@wisc.edu (A.N.J.); rentchler@wisc.edu (E.C.R.); 3Department of Chemistry, University of Wisconsin-Madison, Madison, WI 53706, USA; zli597@wisc.edu (Z.L.); lingjun.li@wisc.edu (L.L.); 4Department of Pathology and Laboratory Medicine, University of Wisconsin-Madison, Madison, WI 53706, USA; pweisman@wisc.edu; 5Carbone Cancer Center, University of Wisconsin-Madison, Madison, WI 53706, USA; 6School of Pharmacy, University of Wisconsin-Madison, Madison, WI 53705, USA

**Keywords:** HGSOC, STICs, precursor lesions, Second Harmonic Generation, collagen remodeling, mass spectrometry

## Abstract

**Simple Summary:**

The collagen architecture in the extracellular matrix (ECM) is highly remodeled in high grade serous ovarian cancer (HGSOC). Many of these tumors begin in the fallopian tubes (FT) before metastasizing to the ovaries and it is important to study ECM alterations in carcinogenesis. Here, we used Second Harmonic Generation (SHG) microscopy to classify changes in the collagen fiber morphology in normal FT, and precursor pure p53 signatures and serous tubal intraepithelial carcinoma (STICs) in tissues with no HGSOC. Using a machine learning approach based on image features, we were able to discriminate the tissue groups with good classification accuracy. We additionally performed mass spectrometry analysis of normal and HGSOC tissues to associate the differential expression of collagen isoforms with fiber morphology alterations. This work provides new insights into ECM remodeling in early stage HGSOC and suggests the combined use of SHG microscopy and mass spectrometry as a new diagnostic/prognostic approach.

**Abstract:**

**Background:** The collagen architecture in high grade serous ovarian cancer (HGSOC) is highly remodeled compared to the normal ovary and the fallopian tubes (FT). We previously used Second Harmonic Generation (SHG) microscopy and machine learning to classify the changes in collagen fiber morphology occurring in serous tubal intraepithelial carcinoma (STIC) lesions that are concurrent with HGSOC. We now extend these studies to examine collagen remodeling in pure p53 signatures, STICs and normal regions in tissues that have no concurrent HGSOC. This is an important distinction as high-grade disease can result in distant collagen changes through a field effect mechanism. **Methods:** We trained a linear discriminant model based on SHG texture and image features as a classifier to discriminate the tissue groups. We additionally performed mass spectrometry analysis of normal and HGSOC tissues to associate the differential expression of collagen isoforms with collagen fiber morphology alterations. **Results:** We quantified the differences in the collagen architecture between normal tissue and the precursors with good classification accuracy. Through proteomic analysis, we identified the downregulation of single α-chains including those for Col I and III, where these results are consistent with our previous SHG-based supramolecular analyses. **Conclusion:** This work provides new insights into ECM remodeling in early ovarian cancer and suggests the combined use of SHG microscopy and mass spectrometry as a new diagnostic/prognostic approach.

## 1. Introduction

High Grade Serous Ovarian Cancer (HGSOC) is a highly metastatic disease, defined genetically by mutations in the tumor suppressor genes Tp53, BRCA I, and BRCA II, and DNA copy number alterations [1]. While these mutations are well-documented, the associated effects in the tumor microenvironment (TME), especially in terms of remodeling of the extracellular matrix (ECM), have not been well-studied. Such modifications occur in essentially all epithelial cancers, and are important in HGSOC because this disease can metastasize while the lesions are smaller than the resolution of clinical imaging modalities (e.g., ultrasound, CT, MRI, and PET) [2,3,4]. 

Serum tests (CA125 and HE4) do not have sufficient specificity and sensitivity for early reliable diagnosis [5,6,7]. As a result of these factors, in more than 70% of patients, HGSOC is detected at an advanced stage when the treatment options are limited. We postulate that the development of efficacious imaging/screening modalities requires a more thorough understanding of the HGSOC microenvironment.

For this purpose, we utilized the high-resolution collagen specific, optical modality of Second Harmonic Generation (SHG) microscopy to probe all levels of collagen structure (molecular through fiber). Importantly, we previously developed machine learning algorithms to differentiate between normal and high-risk ovarian stromal tissues as well as cancer sub-types based on the 3D collagen fiber morphology patterns [8]. We were able to discriminate HGSOC and normal tissues with excellent classification accuracy (~95%) [9] and other sub-types with good accuracy (~85%). 

We have also documented sub-resolution macro/supramolecular changes (protein helix attributes) and fibril organization (size and packing) in the aberrant tissue classes [10,11]. Collectively, these studies showed that the collagen fibers are more aligned in HGSOC than in the corresponding normal tissues or other ovarian cancer sub-types and that the underlying supramolecular and fibril structures are more disordered. These results are consistent with improperly synthesized new collagen and/or faster turnover in normal Collagen I (Col I).

As the majority of HGSOC cases originate through precursors in the fallopian tube (FT) secretory epithelium [12,13,14,15], it is also important to investigate the corresponding collagen alterations as these can be biomarkers for early diagnosis of the disease. Both p53 signatures and serous tubal intraepithelial carcinomas (STICs) have been identified as two early precursors of HGSOC in the FT [16]. Aside from a loss of cilia and outgrowth of secretory cells, p53 signatures are defined by their aberrant and intense p53 staining [15]. 

In contrast, STICs are associated with high p53 intensity and the acquisition of cellular changes, where the morphology becomes more disorganized. Malignant disease in the FT (and primary ovary) that develops from the STICs has additional morphological alterations along with a high proliferative index [1]. Similar to the ovarian TME itself, the corresponding changes in the FT microenvironment, especially in terms of the collagen architecture, are not well-known beyond standard hematoxylin and eosin (H&E) histology, which is not sensitive to detailed fibrillar features.

We recently used SHG microscopy to investigate the collagen fiber structure in concurrent STIC and HGSOC fallopian tube tissues [17]. Interestingly, the collagen morphology in HGSOC resembled that occurring in the ovary itself, and using a multivariate analysis, excellent classification accuracy (~95%) was obtained relative to STICs and normal regions in the same tissue. However, a more modest accuracy (~75%) was obtained between normal and STIC regions, suggesting that more detailed analyses are required to define these collagen structural changes.

Here, we extended these studies to pure p53 and STIC precursors to determine if early changes in the collagen fiber morphology are detectable along with the p53 molecular changes. It is also important to complete these studies in the absence of concurrent HGSOC as these lesions can result in the transformation of distant collagen through a field effect mechanism [18,19,20,21] and obscure the morphologic changes associated only with the precursor states. We also used mass spectrometry to examine the molecular changes that underlie remodeling, e.g., the up- or down-regulation of collagen isoforms. The correlation of differential isoform expression to changes in the collagen fiber morphology has not been previously investigated, and we suggest that this analysis can provide useful insights into remodeling in early disease progression.

## 2. Methods

### 2.1. Archived Human Tissues

In this retrospective study, archived fallopian tubes and ovarian tissues from the University of Wisconsin Carbone Cancer Center Tissue Bank and the University of Wisconsin Department of Surgical Pathology were analyzed under an IRB-approved protocol (protocol #2019-0211). Flash frozen normal (*N* = 3) and tumor fallopian tube tissues *(N* = 3) were analyzed via mass spectrometry, while archived fallopian tube tissues (*N* = 12) were analyzed via SHG microscopy. Appendix A provides additional information on the tissues evaluated.

### 2.2. Sample Processing, Histology, and Mapping of Precursor Lesions

The SEE-FIM protocol [12] was executed to identify fallopian tube samples with HGSOC and HGSOC precursors. Paraffin blocks of the cases with confirmed normal, p53 signatures, STICs, and HGSOC were serially sectioned to obtain 5–10 μm thick sections. The sections were stained with H&E to monitor the morphology and to confirm the presence of HGSOC and its precursor lesions. The slides were also immunostained for p53 using the DO-1 hybridoma (SantaCruz Biotechnology, SantaCruz, CA, USA). Adjacent sections were retained as unstained slides for SHG imaging. The stained slides were examined by a trained pathologist (P.W.) to confirm the diagnosis. This pathological review was used to map the normal tissues, precursor lesions, and HGSOC in the unstained slides.

### 2.3. Sample Processing for Mass Spectrometry-Based Proteomic Analysis

**Protein extraction and digestion.** Each sample was dissolved in 1 mL of extraction buffer (4% sodium dodecyl sulfate, 50 mM Tris-HCl, pH 8) and sonicated using a probe sonicator (Thermo Fisher Scientific, San Jose, CA, USA). Protein extracts were reduced with 10 mM dithiothreitol (DTT) for 30 min at room temperature and alkylated with 50 mM iodoacetamide for another 30 min in the dark before quenching with DTT. The proteins were then precipitated with 80% (*v*/*v*) cold acetone (−20 °C) overnight. The samples were centrifuged at 14,000× *g* for 15 min after which supernatant was discarded.

The pellets were rinsed with cold acetone and air-dried at room temperature. Eight molar urea was added to dissolve the pellets, and 50 mM Tris buffer was used to dilute the samples to a urea concentration <1 M. On-pellet digestion was performed with LysC/trypsin (Promega, Madison, WI, USA) at a 50:1 ratio (protein: enzyme, *w*/*w*) at 37 ℃ overnight. The digestion was quenched with 1% trifluoracetic acid, and the samples were desalted with Sep-Pak C18 cartridges (Waters, Milford, MA, USA). The concentrations of peptide mixture were measured by peptide assay (Thermo Fisher Scientific, San Jose, CA, USA). Ten micrograms of peptide were aliquoted for each sample and dried in vacuo.

**Liquid chromatography (LC)-tandem mass spectrometry analysis.** The samples were analyzed on a Q-Exactive quadrupole Orbitrap mass spectrometer (Thermo Fisher Scientific, San Jose, CA, USA) coupled to a Waters nanoAcquity Ultra Performance LC. Each sample was dissolved in 15 μL 4% acetonitrile (ACN) and 0.1% formic acid (FA) in water before loading onto a 75 μm inner diameter homemade microcapillary column, which was packed with 15 cm of Bridged Ethylene Hybrid C18 particles (1.7 μm, 130 Å, Waters, Milford, MA, USA) and fabricated with an integrated emitter tip. Mobile phase A was composed of water and 0.1% FA, while mobile phase B was composed of ACN and 0.1% FA. 

LC separation was achieved across a 120-min gradient elution of 4% to 30% mobile phase B at a flow rate of 300 nL/min. Survey scans of peptide precursors from 300 to 1500 *m*/*z* were performed at a resolving power of 70,000 with an automatic gain control (AGC) target of 1 × 10^6^ and maximum injection time of 250 ms. The top 15 precursors were then selected for higher-energy collisional dissociation (HCD) fragmentation with a normalized collision energy of 30, an isolation width of 2.0 Da, a resolving power of 17,500, an AGC target of 1 × 10^5^, a maximum injection time of 150 ms, and a lower mass limit of 120 *m*/*z*. The precursors were subject to dynamic exclusion for 45 s with a 10 ppm tolerance. Each sample was acquired in technical triplicates.

**Data analysis.** The raw files were searched against the UniProt *Homo Sapiens* reviewed database (February 2020) using MaxQuant (version 1.5.2.8) [22] with trypsin/P selected as the enzyme and two missed cleavages allowed. Carbamidomethylation of cysteine residues (+57.02146 Da) was chosen as fixed modification and variable modifications included oxidation of methionine residues (+15.99492 Da), acetylation at protein N-terminus (+42.01056 Da), and hydroxylation on proline residues (+15.99492 Da). The “LFQ quantification” and “Match between runs” features were enabled in MaxQuant. Search results were filtered to a 1% false discovery rate (FDR) at both the peptide and protein levels. 

Peptides that were found as reverse or potential contaminant hits were filtered out, and all other parameters were set as the default. ECM proteins were identified and classified by matching the results to Human Matrisome dataset [23]. Proteins were considered as identifiable when detected in at least one sample and quantifiable when detected in at least two samples in each group. Missing intensities were replaced using the “replace missing values from normal distribution” feature in Perseus [24] (version 1.6.0.7) prior to further processing. 

Two-sample Student’s *t* tests with a two-tailed distribution for binary comparison and hierarchical clustering analysis were conducted using Perseus. The volcano plot was generated using R packages. The mass spectrometry proteomics data have been deposited to the ProteomeXchange Consortium via the PRIDE partner repository [25] with the dataset identifier PXD025864.

### 2.4. Second Harmonic Generation (SHG) Microscopy

The SHG laser scanning microscope used here has been described in detail previously and only the salient features are given here [26]. The excitation source was a mode-locked Titanium Sapphire laser (Mira; Coherent, Santa Clara, CA, USA), providing 890 nm excitation and coupled to an upright microscope (BX61; Olympus, Tokyo, Japan). Laser scanning and data acquisition were achieved through home written LabVIEW code and National Instruments FPGA (National Instruments, Austin, TX, USA). The SHG Excitation used a 40 × 0.8 NA water immersion lens (LUMPlanFL/IF; Olympus, Tokyo, Japan) and a 40 × 0.9 NA condenser for collection of the forward propagating signal. The lateral and axial resolutions of the system were approximately 0.7 and 2.5 µm, respectively, as this is sufficient for resolving collagen fibers.

The SHG emission has an associated directionality resulting from the sub-resolution fibril structure [27], and we acquire the forward and backward propagating signals [28]. These respective components were collected using identical photon-counting detectors (7421 GaAsP; Hamamatsu, Hamamatsu City, Japan) with the backward detector in a non-descanned geometry. For each channel, the SHG wavelength (445 nm) was isolated with a dichroic mirror and 10 nm FWHM bandpass filter (Semrock, Rochester, NY, USA). Circular polarization was used for imaging as this state excites all fiber orientations equally. This polarization of the excitation laser was determined at the focus by imaging dye-labelled vesicles [26]. The collected images were 512 × 512 pixels with a field of view of 180 × 180 µm. The image acquisition time was 3 s per frame with three-frame Kalman averaging.

A total of 81 image stacks were analyzed and utilized for classification across three tissue groups: distal normal (*N* = 37), p53 only signature (*N* = 7), and STIC lesion (*N* = 37). We also included a comparison with HGSOC tissues (*N* = 33) from confirmed cancer patients in some of the analyses.

### 2.5. Image Analysis

An initial set of features to be used for linear discriminant analysis (LDA) was generated from the outputs of Gray Level Co-Occurrence Matrix (GLCM) analysis, two-dimensional fast Fourier transform (2D-FFT) methods, and the curvelet transform combined with the FIRE extraction algorithm (CT-FIRE) [29]. FIJI with the Texture Analyzer plugin was used to calculate five GLCM texture features associated with the similarities and differences between adjacent pixels: Angular Second Moment (ASM), Entropy, Inverse Difference Moment (IDM), Contrast, and Correlation. 

The 2D-FFT analysis was performed in FIJI using the Radial Profile Extended and the Oval Profile Plot plugins to characterize the alignment and radial exponential decay of the image power spectrum. We previously described the radial- and azimuthal-averaging procedures used in this approach [17]. All curve fitting of this data was done in Origin 2018 (OriginLab, Northampton, MA, USA). 

CT-FIRE was utilized to perform curvelet transform and fiber extraction to characterize individual fiber morphology features (fiber length, width, and straightness). As there was insufficient signal from the weaker backward channel for the CT-FIRE analysis, only the data readouts from the forward channel were analyzed. Lastly, the image coverage (packing coefficient) was quantified by creating a binary mask over a dynamic lower threshold and calculating the fraction of the resulting non-vanishing pixels. The methods and features used are summarized in Table 1.

The SAS software (SAS Institute Inc., Cary, NC, USA) was used to reduce the full feature set to the most significant metrics via forward selection at a significance level of α = 0.35 (STEPDISC procedure). After the features were selected, the truncated dataset was inputted to a linear discriminant analysis (LDA) performed with singular value decomposition and N-weighted priors. For this portion, 37 crops as normal, 6 as p53 signatures, and 37 as STIC were analyzed. 

The accuracies and F_1_ scores of the trained model were calculated for each class, where the latter quantity is the harmonic mean of precision (TP/(TP + FP)) and recall (TP/(TP + FN)), where TP, FP, and FN are true positives, false positives, and false negatives, respectively. An F_1_ score is an additional metric of classifier accuracy which accounts for class imbalance. Receiver Operating Characteristic (ROC) curves were generated under five-fold cross-validation to assess the binary classifier performance and to quantify the trade-off between the true positive rate (TPR) and false positive rate (FPR).

## 3. Results

### 3.1. Mass Spectrometry Analysis

We explored the changes in ECM and ECM-related proteins, with a focus on collagen isoform expression in human normal ovarian and tumor tissues via mass spectrometry-based proteomics approaches. In total, we identified 233 ECM proteins, where 25 were single α-chains comprising several collagen isoforms. We observed numerous proteins that were only found in either tumor or normal samples (Appendix A). For instance, COL8A1, COL8A2, and COL21A1 were only detected in normal tissues whereas COL7A1 and MUC1 were exclusively identified in tumor samples (Appendix A). Importantly, even with a low number of replicates, we were able to identify and quantify many differentially expressed (especially down-regulated) ECM proteins (Appendix A) in the tumor group.

As shown in the cluster map in Figure 1, 15 single α-chains from different collagen isoforms were present in all samples, and statistical proteomic landscape differences were found between the two groups. Interestingly, we observed decreased expression levels of many of these chains in the tumor samples (e.g., those for COL1, COL3, COL5, COL6, COL12, and COL14). Moreover, differences in multiple single chains of the same collagen isoform were detected (e.g., COL1A1 and COL1A2), improving the confidence of our observations. These findings support the existence of unique matrisome features in each group, where there were larger intergroup differences than intragroup variations. This was also borne out by analysis of the principal components (not shown). A full list of identified and quantified proteins can be found in the Appendix A.

### 3.2. SHG Imaging and Analysis

**Locating and mapping pure p53 signatures and STIC lesions.** To obtain tissues with p53 signatures and STIC lesions with no concurrent tumors, we focused on FT tissues obtained from gynecologic surgeries not related to HGSOC. An archival text-based survey of patients with STIC over a 5-year period (2013–2018) revealed 12 patient cases that met our criteria. Only 2 of the 12 had concurrent p53 signature lesions along with the STIC. None of these cases had any pathological characteristics for HGSOC. The low number of cases with pure p53 signature and STIC lesions is consistent with their reported low incidence in these cohorts.

To confirm that these were pure precursors without the presence of cancer, routine histological stains (H&E and p53) were completed. Figure 2 provides an example of one such precursor along with an example of HGSOC. In addition to the normal H&E distribution, the weak p53 reactivity is consistent with the absence of cancer.

The fallopian tube tissue slices for SHG imaging were unstained and, therefore, posed a challenge to accurately identify the normal regions, p53 signatures, and STICs. This issue was addressed by the H&E and p53 staining of an adjacent section of the same tissue, where these slides were used as a template to manually map and score the normal areas and precursors on the slides used for SHG imaging. This workflow was completed for all FT samples and is outlined in Figure 3. 

A trained gynecologic pathologist (P.W.) first identified the specific areas of normal tissue (green rectangle), p53 signature (red rectangle), and STIC (blue rectangle) (Figure 3a). For orientation purposes, bright field images at 4×, 20×, and 40× at each spot were also taken (not shown). The correlated unstained tissue (Figure 3b) was mapped according to the designated areas and imaged. The SHG images of collagen in areas corresponding to normal tissue, the p53 signature, and STIC are shown in Figure 3c, followed by the corresponding CT-FIRE and 2D-FFT analysis, which is quantified in Figure 4.

Our previous study indicated that a collagen coverage of less than 70% significantly altered the accuracy of the image analysis techniques, and many of these regions had sparser coverage. As a solution, representative image stacks for each channel were duplicated and cropped to 45 × 45 microns field of view (FOV), and we selected regions of interest (ROIs) with sufficient coverage. The p53 signatures and STIC lesions were small in spatial extent and localized in their respective tissue and yielded z-stacks comprised of 10 or fewer optical sections. Cropped images of each group (normal, *N* = 37; p53 signature, *N* = 6; STIC, *N* = 37) were analyzed using the image analysis protocols from our previous study [17].

**Texture and other image features.** For our analysis, we included features from GLCM, 2D-FFT methods, and CT-FIRE, where these techniques were applied to both the forward and backward channels. Unlike fluorescence where the emission is isotropic, SHG has an emission directionality that is related to the underlying structure [27]. Specifically, smaller and more disorganized features can appear in the backward channel, where these are often obscured in the forward collected signal. We have shown that these images are sufficiently different through the structural similarity index [17] to justify the inclusion of both signal pathways as independent features. However, the backward signal is intrinsically weaker, and insufficient signal was present for use in the CT-FIRE analysis. The data from all these analyses are summarized in Figure 4.

Although we are specifically focused on discrimination between the distal normal and precursors (p53 and STIC), we included HGSOC as a point of comparison. In good agreement with a previous study [17], we found that images from HGSOC were associated with higher entropy and correlation, as well as lower contrast with respect to other groups (Figure 4a). In this context, lower contrast refers to the similarity of pairwise pixels rather than low signal and is also consistent with high correlation. However, we did not find any significant differences in the GLCM metrics between either of the precursors and distal normal regions. 

2D-FFT methods were able to distinguish HGSOC as having higher alignment in the forward and backward channels, where both readouts of this metric should trend in the same direction. There was also a greater relative occurrence of high frequency (smaller) features in HGSOC, which is indicated by the higher time constant in its radial power spectrum (Figure 4b). Conversely, the STIC and p53 signature groups were characterized by lower alignment in the forward channel, although this was not significantly different. Lastly, while HGSOC was associated with straighter fibers, no differences were found between the two precursors and normal (Figure 4c).

**Linear Discriminant Analysis (LDA).** While none of the individual metrics from Figure 4 showed differences between the two precursors, we can attempt to obtain discrimination through the development of a linear discriminant (LD) model. This process can provide improved classification even if the individual components are not themselves statistically different, and we have used this process previously [17,30]. Since the collagen morphology of HGSOC in the FT is markedly distinct and already characterized [8,17], we limited our discrimination analysis to the two precursors and distal normal regions.

In order to better differentiate between the precursor groups (and to prevent overfitting in the trained LD model), the feature space generated by the GLCM, 2D-FFT methods, CT-FIRE, and the packing coefficient was limited via forward selection up to a significance level of α = 0.35 (SAS/STEPDISC procedure). The most significant variables for discriminating between these groups (Packing Coefficient [B], ASM [B], IDM [F], Entropy [F], Entropy [B], Alignment [F], Alignment [B], Time Constant [B], and Fiber Straightness) were then used to construct canonical variables and visualize the classes via a scatter plot (Figure 5).

These forward selected metrics were then used to train an LD model capable of distinguishing between the three tissue groups by a set of binary classifiers (One-vs-Rest or OvR). Through this analysis, we were able to achieve accuracies and F_1_ scores between ~65 to 91% and 9.1–66.0, respectively (Table 2). In particular, we achieved good discrimination for the distal normal and STIC lesion groups, despite low sample sizes (*N* = 37 each). The corresponding AUROCs for these classifiers were somewhat low (0.71 and 0.62 for the distal normal and STIC lesion, respectively; see Figure 6) but this may be improved upon by increasing the size of the training set.

Despite the high overall accuracy (~91%) for p53 signature classification, the corresponding F_1_ score of 9.1 indicates a low number of true positives. To overcome the limitation of low *N* for the p53 signature group, we trained a classifier to distinguish between the distal normal and precursor regions, where the p53 signature and STIC were aggregated into a more general precursor group. Through a similar model using slightly different metrics, we achieved a high accuracy and F_1_ score (74.7 and 77.8, respectively), as well as an AUROC of 0.68. The scatter plot and ROC curve for this model are included in the Appendix A.

## 4. Discussion

Using SHG microscopy, we have previously shown that there are significant changes in the collagen fibrillar morphology in HGSOC in the ovary itself, as well as in the fallopian tubes [8,17]. More subtle differences were observed in STIC regions that were co-existent in tissues with HGSOC. It is important to examine the collagen morphology in pure precursor tissues (p53 and STIC) to determine when characteristic collagen fiber alterations can be detected by SHG, as these could be used as unique diagnostic biomarkers of early-stage disease. 

It is further important to perform these investigations in tissues without HGSOC, as these lesions can induce collagen remodeling in distant regions through a field effect mechanism [19]. However, the acquisition of these pure precursors is clinically rare and it took extensive time and effort to identify even the relatively low number of suitable banked tissues used here. Specifically, an experienced gyn/onc pathologist (P.W.) scanned patient cases from over a 5-year period for this study.

We also sought to determine if there were differences in the collagen expression patterns between normal ovarian tissues and HGSOC, and further, if these were related to the collagen morphology changes visualized by SHG microscopy. This is important as, while previous studies have suggested the up-regulation of several collagen isoforms in HGSOC (e.g., Col III and VI) [31,32], these studies used immunostaining and the results were not verified by quantitative molecular techniques. 

This is potentially problematic as most available antibodies lack a high level of specificity for different isoforms (specifically Col I vs. Col III) [33] as the same epitope at the end of the helix is often tagged. As a consequence, the in vivo isoform composition in HGSOC is not yet definitively known and this represents a large gap in knowledge. We note that we have used self-assembled in vitro models of known composition [33,34] to show how the incorporation of Col III and Col V affects the fibrillar structure, but it has not yet been possible to create a direct link between collagen proteomic changes and fiber alterations in tumors. This is because SHG is not sensitive to non-collagen components, and more generally applicable techniques, such as mass spectrometry (MS), are required for this purpose.

To begin making this connection, we utilized MS and found the decreased expression of numerous single α-chains from several different collagen isoforms in HGSOC. Of particular note, the expression of Col I and Col III chains were both downregulated, where the latter is not consistent with previous immunostaining data [31]. This is likely due to the qualitative nature of immunostaining and the lack of specificity of the available Col III antibodies. 

Interestingly, we previously used detailed SHG analysis to show that there were large structural changes in the α-chains; however, these were not consistent with an increase in Col III expression [11]. Moreover, the triple helical structure was found to be more disorganized in HGSOC. This disorganization was also validated by our wavelength dependent optical scattering measurements, which probed size scales over the range of ~50 nm–1μm [30]. Collectively, the MS data and our previous macro/supramolecular SHG analyses suggest there may be transcriptomic and/or post-translational modifications in HGSOC coinciding with characteristic, pronounced changes in the collagen fiber morphology.

We did not expand the proteomic analysis to the p53 and STIC precursors as the tissues were insufficient in volume. However, given the current paradigm in HGSOC that the ovary is the metastatic site from the FT [16,35], the genetic modifications giving rise to different isoform expression are expected to be similar in both sites. Indeed, we showed that the collagen fiber morphology in HGSOC was highly similar in the FT and ovary [17], suggesting that similar proteomic modifications occur in both tissues. 

Given the congruence in the SHG and the proteomic data suggesting differential collagen isoform expression in normal and HGSOC tissues, these observations further support the validity of performing the MS analysis on more available ovarian stromal tissues. We further suggest that the SHG analysis of the collagen in precursors is a true reflection of the biochemical alterations occurring during carcinogenesis and is an important surrogate measurement that allows for non-destructive analysis, preserving the rare precursors for IHC and transcriptomic analysis.

Based on our prior work [17], we expected that the collagen organization differences between the two precursors (p53 only and STICs) would be less pronounced relative to that of high grade disease. However, our linear discriminant model still achieved good accuracies, F_1_ scores, and ROC curves, which were further improved by grouping p53 signatures and STIC lesions together. Importantly, even with the small sample size, our analysis supports our hypothesis that collagen alterations in the FT occur prior to frank HGSOC. 

We note that, since the collagen alterations of early-stage disease are subtle, the classifier performance will be sensitive to under-sampling. We suggest that with a much larger specimen number, the performance could be improved to a standard that is suitable for future clinical applications. The sample thickness and coverage and limited collagen density leading to relatively weak SHG signal intensities were major limitations in this study. We suggest that higher accuracies should be achievable on thicker sections (~50–100 μm), which are readily imaged by SHG.

Another limitation of the study at the moment is that there is no clear definition of the biochemical factors that affect the change in the collagen structure in HGSOC or in precursor lesions. The differential expression of collagen isoforms combined with specific post-translational processing may contribute to the change in collagen structure. The successful implementation of mass spectroscopy and transcriptomic analysis will be necessary to obtain the biochemical understanding of changes occurring in the collagen and the surrounding ECM. For example, other markers, such as fibronectin, laminin, and secreted MMPs, have been suggested to have altered regulation in HGSOC and could be added to the analysis.

We foresee both long and short-term applications based on our observations and methodology. For the former, it may be possible to construct an SHG laser-scanning micro-endoscope to be used in conjunction with laparoscopy or hysteroscopy [36,37]. Analogous fiber optic-based scanning approaches are currently under development for other pathologies [38,39], and these should be adaptable for FT imaging. 

The scheme may be feasible for the in vivo detection of precursor lesions, especially since the majority of the significant variables for class discrimination came from the SHG backward channel, which would be the usable direction in an endoscopic configuration. In the shorter term, the ex vivo analysis of resected fallopian tube tissue from risk reduction surgery can be used either as a pre-screen or to complement the histology and also to identify MSbased proteomic correlations.

## 5. Conclusions

The collagen fiber morphology is highly remodeled in HGSOC in the ovary or the fallopian tubes, where the fibers become more aligned relative to normal tissues and other tumor sub-types. However, for clinical applications, it is important to investigate these alterations in the precursors (p53 and STICs) due to the early metastasis of HGSOC. Unfortunately, there is a limited availability of pure precursor lesions without the presence of malignant tissue. 

Still, due to the high specificity and sensitivity to collagen morphology afforded by SHG microscopy, sufficient discrimination between distal normal regions and p53 and STIC precursors was attained in this limited study to demonstrate proof of concept. Moreover, mass spectrometry analyses showed concurrent proteomic changes in normal and HGSOC tissues, where many collagen single α-chains were downregulated. These results suggest that the combined use of MS proteomic and SHG microscopy analyses forms a basis for further in vivo and ex vivo explorations of HGSOC and its precursor lesions.

## Figures and Tables

**Figure 1 cancers-13-02794-f001:**
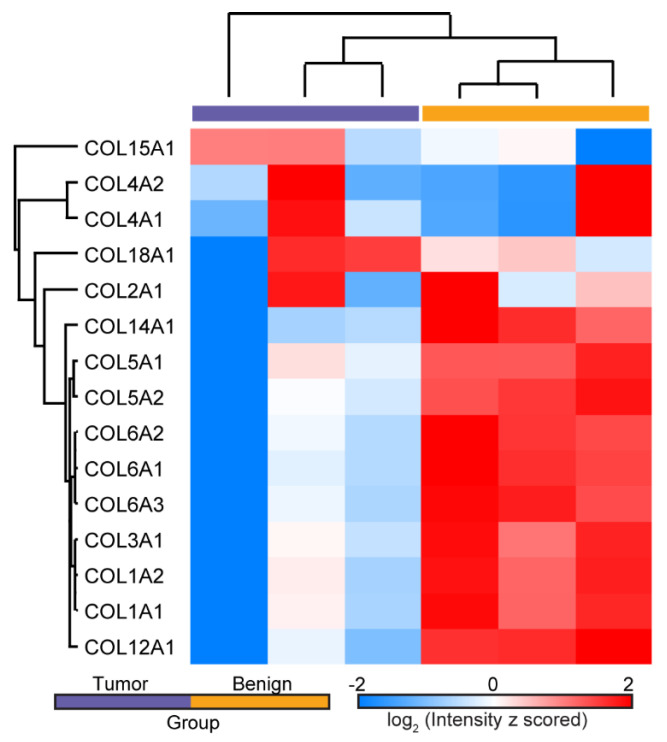
Hierarchical clustering of LFQ ion intensities showing the differential expression of single α-chains from several collagen isoforms in normal ovarian and HGSOC tissues. Several chains were down-regulated in HGSOC.

**Figure 2 cancers-13-02794-f002:**
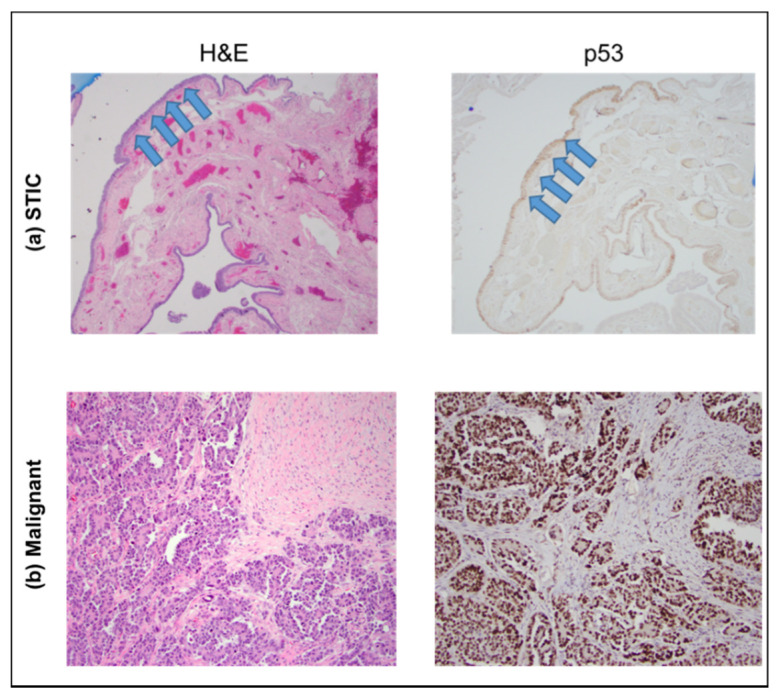
H&E (**left**) and p53 (**right**) staining of sections showing a STIC lesion (top, (**a**) images) and HGSOC (bottom, (**b**) images), respectively. The STIC lesion is confined to the fallopian tube epithelial surface (arrows), whereas the HGSOC forms a destructive, expansile lesion within the stroma of the involved tissue. The images were acquired at 40×.

**Figure 3 cancers-13-02794-f003:**
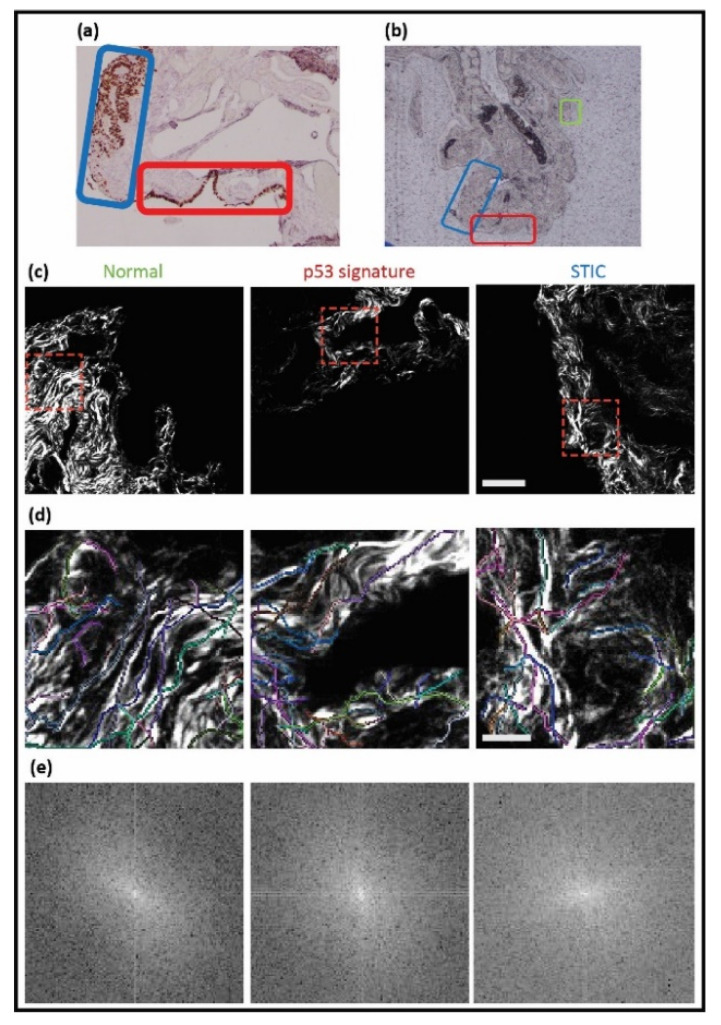
Workflow for identifying p53 and STIC precursors. A pathologist defined these tissues in mutant p53 stained images ((**a**), 4×), where green, red, and blue correspond to normal regions, p53, and STIC, respectively. The correlated unstained regions ((**b**), 4×) were identified and then used for SHG imaging, where the corresponding images are shown in (**c**) (field size = 180 × 180 microns). (**d**,**e**) CT-FIRE overlays and 2D-FFT power spectra of the cropped SHG images (red-dashed boxes) are shown.

**Figure 4 cancers-13-02794-f004:**
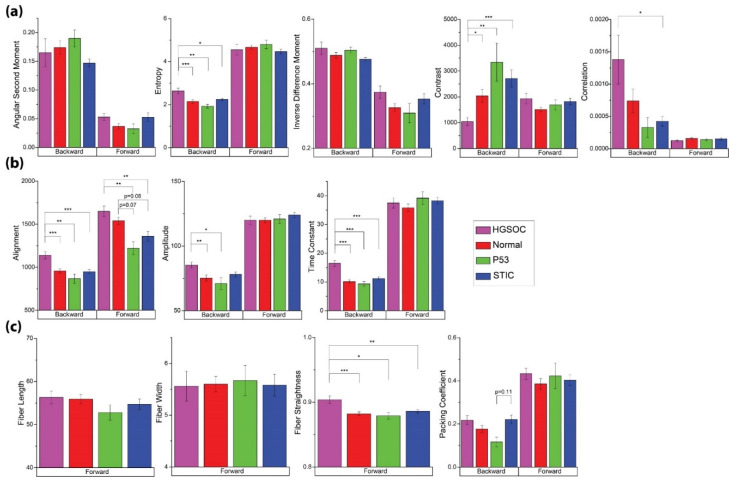
Image and texture features from the analyses given in Table 1 for the four tissue classes of normal, HGSOC, p53, and STIC. (**a**) GLCM features. (**b**) 2D-FFT parameters. (**c**) CT-FIRE outputs and Packing Coefficient. Significant *p*-values are denoted as follows: * = *p* < 0.05, ** = *p* < 0.01, and *** = *p* < 0.001.

**Figure 5 cancers-13-02794-f005:**
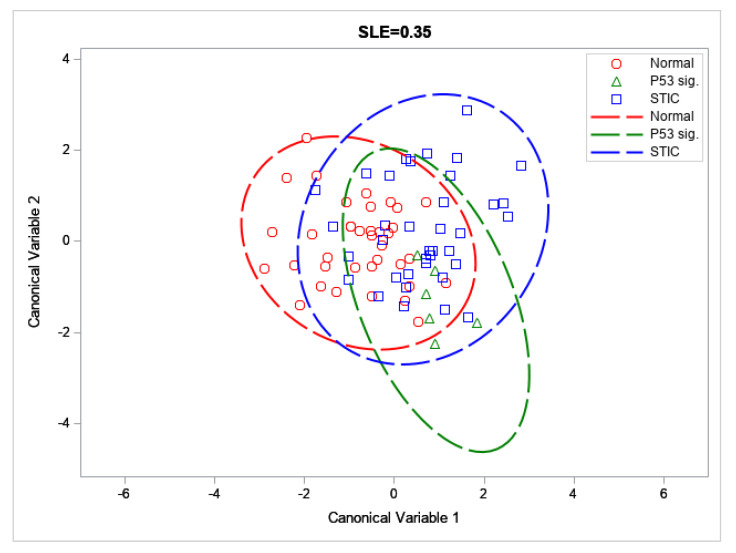
Scatter plot of the distal normal, p53 signature, and STIC across the first two canonical variables; with 95% confidence ellipses depicted by dashed lines.

**Figure 6 cancers-13-02794-f006:**
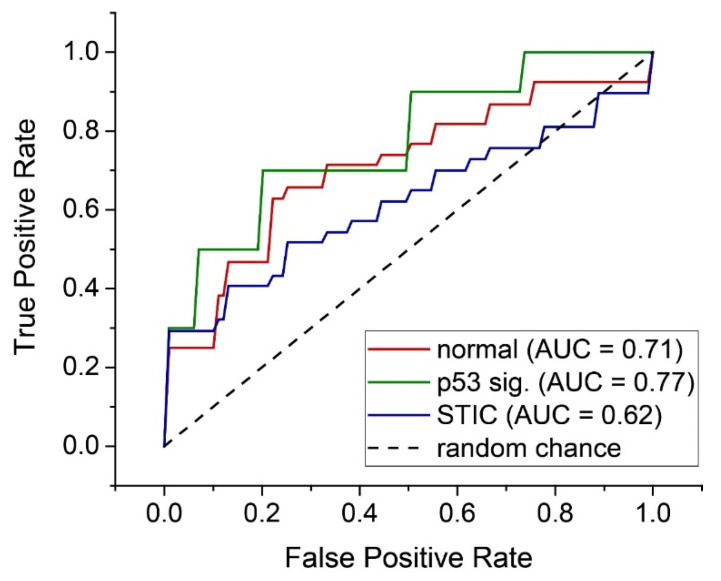
Individual ROC curves for each OvR classifier for the three tissue classes.

**Table 1 cancers-13-02794-t001:** Summary of different metrics used for Linear Discriminant Analysis (LDA).

Method	Feature	Measurement Output
GLCM—encodes the intensity differences of adjacent pixel pairs into a matrix	ASM	Homogeneity of image texture
	Entropy	Randomness of image intensity distribution; inversely correlated to ASM
	IDM	Homogeneity of image intensity
	Contrast	Local intensity variations; inversely correlated to IDM
	Correlation	Correlation of intensity between adjacent pixels
2D-FFT—converts an image into its power spectrum prior to analysis	Alignment	Overall alignment of image by examining signal at different spatial frequencies
	Amplitude	Signal strength at lowest spatial frequencies
	Time Constant (TC)	Time constant of radial power spectrum, inversely proportional to exponential decay rate (k) and related to the incidence of high frequency features
CT-FIRE—identifies discrete fibers in a 2D image	Fiber length	Average length of fibers in image
	Fiber width	Average width of fibers in image
	Fiber straightness	Average straightness of fibers in image
Packing coefficient	Image coverage; related to the density of the collagen network in the image	

**Table 2 cancers-13-02794-t002:** The accuracies and F_1_ scores for the OvR classifier.

Group	Accuracy %	F_1_ Score
Distal normal	69.6	66.0
p53 signature	90.8	9.1
STIC lesion	65.4	61.0

## Data Availability

The mass spectrometry proteomics data have been deposited to the ProteomeXchange Consortium via the PRIDE partner repository with the dataset identifier PXD025864.

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
