# Peer review of "Evaluation of Collagen Alterations in Early Precursor Lesions of High Grade Serous Ovarian Cancer by Second Harmonic Generation Microscopy and Mass Spectrometry"

_cancers, 2021, doi:10.3390/cancers13112794_

Round 1

Reviewer 1 Report

The manuscript from Gant and colleagues presents an innovative strategy to evaluate the collagen -ECM remodelling associated to the onset of high grade serous ovarian cancer. From the technical perspective the Authors chose a state-of-the-art combination of Second Harmonic Generation Microscopy with Proteomics (Mass Spectrometry based). The Authors previously published the application of SHG microscopy for similar purposes, hence the value of the paper contributes to the consolidation of their expertise in the field.

Overall, the manuscript is valuable and provides a novel angle for the interpretation of an important adaptive aspect of the ovarian tumor microenvironment, however some points prevent its publication in the present format and require revision:

-Formal organization of the manuscript: The Authors describe 6 figures, however in the version of the manuscript received by the reviewer only 5 figures were included! Moreover, figure legends were mismatching the graphics, hampering the readability. Figure style and lettering do not appear homogeneous throughout the manuscript and should be corrected.

Figure 2 does not present any detail about image magnification / scale bars.

-Image analysis features (Figure 4?): this is a core information for the manuscript. At least for the parameters that are significantly regulated, respective images should be included in the figures. One of the final goals of the Authors (also according to the discussion) would be a broader and broader application of their methods in the clinics, hence reference images would greatly enhance the interpretation of other specimens and help other researchers to reproduce their approach. In this respect Table 1 helps, however it might be difficult to visualize how does these parameters correlate with the actual data. Reformatting of Table I is also recommended.  

-Mass spectrometry data: Authors should provide Principal Component Analysis of the data. Moreover, it is common for proteomics analysis to make datasets publicly available via online repository or, in alternative, to provide an overview of the regulated proteins as supplementary datasets. This would greatly support the impact of the paper and it is to be expected for the publication in a high-ranking journal. If there are limiting factors in this respect these should be motivated and if the data presented in the paper belong to a previously analysed/published dataset, this should be clearly stated as well. According to the discussion of the results, it is not clear how changes in proteome profile targeting collagen are positioning with respect to the total proteome alterations. Are the changes in the ECM the only appreciable differences between the samples? How many proteins were regulated in total?

-Discussion; lines 381-384. The Authors identify a discrepancy in the data collected with different methods, at least some hypothesis should be formulated to explain this aspect.

Reviewer 2 Report

The authors used Second Harmonic Generation (SHG) microscopy and machine learning to assess changes in collagen morphology in ovarian cancers with p53 mutations. The authors also performed mass spectrometry analysis to associate differential expression of collagen isoforms with collagen alterations. The proteomic analysis indicated the downregulation of single α-chains in Col I and III. Overall, data indicates that the combined use of microscopy and mass spectrometry may be used as a new diagnostic/prognostic approach. I think that the study is very useful for pathology services as it represents a new diagnostic method. The study is comprehensive and well-designed. However, there are several problems to address. 1. Sub-section titles should be accented. For instance, “Locating and mapping pure p53 signatures and STIC lesions” ( line 246) and “ Linear Discriminant Analysis (LDA)” ( line 317Figure 4:) – should be in bold or Italic font. 2. Line 79: you do not need “To this end”, delete the phrase. Keep academic style. 3.Figure legends were mixed up during pdf file formation. Authors should be careful when submitting pdf file and always verify the document template before submission. 4. Figure 4: specify “ four tissue classes” in the figure legend. 5. why fig.6 represents only 3 tissue classes instead of 4? 6. if you have identified MUC1 only in tumor tissues, why you have not used MUC1 in further analysis? It is unclear. It seems that MUC1 is a promising ovarian cancer marker. 7. Authors should describe the limitation of this study. 8. Lots of mistakes with citation style. Many [ref] were placed behind commas and full stops. All mistakes should be corrected. 9.Last sentence (lines 442-443) is too general; avoid vacuous generalizations.

Author Response

We thank the Reviewer for the detailed comments and suggestions. We addressed all your points and the changes in the manuscript are in red with the line numbers called out.

The authors used Second Harmonic Generation (SHG) microscopy and machine learning to assess changes in collagen morphology in ovarian cancers with p53 mutations. The authors also performed mass spectrometry analysis to associate differential expression of collagen isoforms with collagen alterations. The proteomic analysis indicated the downregulation of single α-chains in Col I and III. Overall, data indicates that the combined use of microscopy and mass spectrometry may be used as a new diagnostic/prognostic approach. I think that the study is very useful for pathology services as it represents a new diagnostic method. The study is comprehensive and well-designed. However, there are several problems to address.

  1. Sub-section titles should be accented. For instance, “Locating and mapping pure p53 signatures and STIC lesions” ( line 246) and “ Linear Discriminant Analysis (LDA)” ( line 317Figure 4:) – should be in bold or Italic font.

They were in bold in our word version. This was a failure of the journal’s pdf conversion (along with the figure caption placing). Still, we have now made the sub-section styles consistent in the methods and results.

  1. Line 79: you do not need “To this end”, delete the phrase. Keep academic style.

We have deleted this.

3.Figure legends were mixed up during pdf file formation. Authors should be careful when submitting pdf file and always verify the document template before submission.

The online pdf convertor greatly altered the word document we had submitted. Despite numerous tries, we could not get the conversion to be successful.

  1. Figure 4: specify “ four tissue classes” in the figure legend.

We have added this.

  1. why fig.6 represents only 3 tissue classes instead of 4?

As noted on line 337 the high grade data was previously characterized before and found to have large separation from the other tissue classes. Here we wanted to optimize our analysis and focus on achieving the best classification accuracy between the precursors.

  1. if you have identified MUC1 only in tumor tissues, why you have not used MUC1 in further analysis? It is unclear. It seems that MUC1 is a promising ovarian cancer marker.

We agree that the selective expression of MUC1 in tumor tissues and not in benign tissues is interesting and has the potential of being included in a future panel of diagnostic markers for detection of the serous ovarian tumors or its precursor lesions. However, we respectfully point out that as indicated in Supplementary Table 1, there are also several other proteins that were expressed exclusively in either the tumor or the benign lesions. Therefore, in future studies we will be investigating multiple markers listed in this table in addition to MUC1.

More importantly, we would like to point out that in addition to demonstrating the differential expression of proteins in tumor and benign tissues by mass spectrometry, the other major goal of this manuscript was to study the difference in collagen structure in the two precursor lesions of high grade serous ovarian cancer. Since collagen structure, to our knowledge, has not been shown to be influenced by MUC1 or other mucins, we do not believe that providing data on MUC1 will add to the information provided in this manuscript. Furthermore, there are strict protocols that are followed by pathologists to identify the p53 signature and STIC precursor lesions. These strict criteria include morphologic analysis and p53 expression patterns of paraffin embedded fallopian tube tissues. These criteria do not include analysis of MUC1 for clinical definition of the precursor lesions and therefore this mucin biomarker was not included in this study.  

  1. Authors should describe the limitation of this study.

We have now added the following paragraph to further highlight a limitation of the study.

“Another limitation of the study at the moment is that there is no clear definition of the biochemical factors that affect the change in the collagen structure in HGSOC or is precursor lesions. Differential expression of collagen isoforms combined with specific post-translational processing may contribute to the change in collagen structure. The successful implementation of mass spectroscopy and transcriptomic analysis will be necessary to obtain the biochemical understanding of changes occurring in the collagen and the surrounding extracellular matrix.”  This has been added on line 437.

In other sections of the Discussion, we have mentioned the lack of sufficient number of samples and collagen coverage and their effect on the analysis shown in the current manuscript.

  1. Lots of mistakes with citation style. Many [ref] were placed behind commas and full stops. All mistakes should be corrected.

These were corrected throughout and are consistent.

9.Last sentence (lines 442-443) is too general; avoid vacuous generalizations.

This was modified to “These results suggest that the combined use of MS proteomic and SHG microscopy analyses forms a basis for further in vivo and ex vivo explorations of HGSOC and its precursor lesions.” (line 469).

Reviewer 3 Report

The authors present the quantification of collagen in p53 signature and STICs lesions. The work is very interesting and promising for the development of diagnostic/prognostic approaches for HGSOC. The paper is clear and well written.

Minor revisions:

Do you think that other ECM markers could be associated to collagen as a pool of markers to be considered for an early diagnosis of HGSOC? Please discuss this aspect in the paper.

Author Response

Yes, other markers such as fibronectin, laminin and secreted MMPs have been suggested to have altered regulation in HGSOC and could be added to the analysis. This has been added to line 437.

Round 2

Reviewer 1 Report

The revision work was compliant to the requests of the reviewer.

Author Response

We have corrected minor formatting issues.

Reviewer 2 Report

I am satisfied with the revised version of the manuscript. Authors provided responses to all my questions/suggestions. However, authors still make mistakes with citations/referencing. Authors did not leave space between words and [citation]; example - ."..vesicles[26]". Authors should pay attention to these small mistakes and correct them all.

Author Response

We have corrected the minor formatting issues on the references.